# Improvement in Fatigue Strength of Chromium–Nickel Austenitic Stainless Steels via Diamond Burnishing and Subsequent Low-Temperature Gas Nitriding

Jordan Maximov [1],*, Galya Duncheva [1], Angel Anchev [1], Vladimir Dunchev [1] and Yaroslav Argirov [2]

1   Department of Material Science and Mechanics of Materials, Technical University of Gabrovo,
    5300 Gabrovo, Bulgaria; duncheva@tugab.bg (G.D.); anchev@tugab.bg (A.A.); v.dunchev@tugab.bg (V.D.)
2   Department of Material Sciences, Technical University of Varna, 9010 Varna, Bulgaria; jaroslav.1955@abv.bg
*   Correspondence: jordanmaximov@gmail.com

**Abstract:** Chromium–nickel austenitic stainless steels are widely used due to their high corrosion resistance, good weldability and deformability. To some extent, their application is limited by their mechanical characteristics. As a result of their austenitic structure, increasing the static and dynamic strength of the components can be achieved by surface cold work. Due to the tendency of these steels to undergo intercrystalline corrosion, another approach to improving their mechanical characteristics is the use of low-temperature thermo-chemical diffusion processes. This article proposes a new combined process based on sequentially applied diamond burnishing (DB) and low-temperature gas nitriding (LTGN) to optimally improve the fatigue strength of 304 steel. The essence of the proposed approach is to combine the advantages of the two processes (DB and LTGN) to create a zone of residual compressive stresses in the surface and subsurface layers—the enormous surface residual stresses (axial and hoop) introduced by LTGN, with the significant depth of the compressive zone characteristic of static surface cold working processes. DB (both smoothing and single-pass hardening), in combination with LTGN, achieves a fatigue limit of 600 MPa, an improvement of 36.4% compared to untreated specimens. Individually, smoothing DB, single-pass DB and LTGN achieve 540 MPa, 580 MPa and 580 MPa, respectively. It was found that as the degree of plastic deformation of the surface layer introduced by DB increases, the content of the S-phase in the nitrogen-rich layer formed by LTGN decreases, with a resultant increased content of the $\varepsilon$-phase and a new (also hard) phase: stabilized nitrogen-bearing martensite.

**Keywords:** chromium–nickel austenitic stainless steels; surface cold working; low-temperature nitriding; residual stresses; fatigue strength

## 1. Introduction

Chromium–nickel austenitic stainless steels (CNASS) are widely used due to their high corrosion resistance, good weldability and deformability in both hot and cold conditions [1]. To some extent, their application is limited by insufficient hardness, strength and wear resistance. Due to their austenitic structure, increasing the static and dynamic strength of the components can be achieved by surface cold working (SCW). If the nickel content in the steel is below 15 wt% and the degree of cold plastic deformation is sufficiently large, a phase transformation $(\gamma \rightarrow \alpha)$ is observed, i.e., the so-called strain-induced $\alpha'-$martensite is formed, the hardness of which is greater than that of austenite. Thus, a surface layer with significant hardness and residual compressive stresses is formed [2,3].

The increase in the fatigue strength of CNASS is mainly explained by the introduction of compressive residual stresses in the surface and subsurface layers and a significant increase in surface microhardness. Such a modification of the surface layers, in some cases, leads to an increase in the sliding wear resistance [4]. Therefore, increasing the fatigue

strength through a suitable finishing method can, in some cases, lead to improved wear resistance. The object of the present study is to increase the fatigue strength of CNASS.

The SCW methods are static and dynamic. Static methods, often called burnishing methods, are suitable for rotary surfaces but also apply to planar surfaces. Dynamic methods are suitable for more complex surfaces, but the resulting roughness is usually greater than that provided by static methods. In the present study, a static method with sliding friction contact between a diamond-deforming element and the treated surface—diamond burnishing (DB)—is used [5].

In recent years, many studies have been devoted to improving the surface integrity (SI) and operating behavior (wear, corrosion and fatigue resistance) of CNASS components by means of static SCW methods: ball burnishing [6–21], roller burnishing [6,22–25] and DB [2–4,26–35]. There are relatively few studies devoted to improvement in the fatigue behavior of CNASS components by means of SCW [2–4,8–13,18,23]. The most commonly used processes are hardening DB [2–4] and deep rolling [8–13], implemented by hydrostatic ball burnishing. Juijerm and Altenberger [8] increased the rotating bending fatigue strength ($10^6$ cycles) of 304 CNASS by approximately 18% using the deep rolling process. Maximov et al. [2] reported an increase of approximately 37% in the rotating fatigue limit ($10^7$ cycles fatigue strength) for the same grade of steel (304) via five-pass hardening DB. An increase of approximately 39% in the rotating fatigue limit of 316 CNASS was achieved by Maximov et al. [4] using four-pass DB. It is important to note that the information cited above regarding the effect of SCW refers to CNASS obtained in the form of hot-rolled bars, i.e., with initial hot-mechanical strengthening, compared to those without initial heat treatment [3].

Due to the tendency of CNASS to suffer intercrystalline corrosion, another approach to increasing their hardness, wear resistance and strength, as well as reducing fatigue, is the use of low-temperature thermo-chemical diffusion processes (nitriding and/or carburizing). The modification of the surface layers during these processes is a consequence of the retention of the nitrogen and/or carbon atoms in the solid solution of the austenite, resulting in the formation of a supersaturated surface phase, the so-called S-phase [36]. Detailed and systematic information about the effect of this kind of heat treatment process on the SI (including the formation, characteristics and properties of the S-phase) and operating behavior of CNASS is contained in the remarkable review papers [36,37]. Since our research is focused on improving the fatigue behavior of CNASS, the literature survey is limited to the effect of low-temperature nitriding/carburizing (LTN/C) on fatigue strength. Hoshiyama et al. [38] reported an increase of approximately 17% in the rotating fatigue strength of 304 CNASS using LTN (compared with that of untreated steel). In another study by Hoshiyama et al. [39] on the same class of steel, an increase in the rotating fatigue limit of almost 39% is reported. Ceschini and Minak [40] reported an increase of approximately 40% in the rotating fatigue limit of 316L CNASS using LTC. For the same grade of steel subjected to LTC, Peng et al. [41] reported an increase in the fatigue limit of 22% obtained by the tension-compression fatigue test. Stinville et al. [42] studied the low-cycle fatigue behavior of low-temperature, nitrided 316L specimens subjected to symmetric tension-compression testing with controlled plastic strain. Comparing the behavior of untreated specimens and those low-temperature, nitrided specimens for eight hours, these authors have established that an increase in the total stress amplitude during the whole cyclic deformation test appears. Moreover, it is reported that LTN and LTC also significantly improve the fretting-fatigue [43] and corrosion-fatigue [44] properties of CNASS components.

The literature survey shows that the increase in rotating-fatigue strength of CNASS specimens via LTN and LTC is in the range of (22–40)%, commensurate with the increase of (18–39)% achieved by static SCW processes. On this basis, the idea of combining the two approaches (static SCW with LTN/LTC) in order to achieve a synergistic effect in terms of fatigue strength improvement seems promising. In order to eliminate possible misunderstandings of a terminological nature, two concepts will be defined. Processes in which influences of different natures are applied sequentially to the treated surface

in order to achieve a synergistic effect regarding the improvement of SI and operational behavior can be called combined processes. With this approach, the evolution of SI can be studied in correlation with the applied impacts, and on this basis, these impacts can be managed appropriately. If the same effect is sought to be achieved but the impacts are applied simultaneously, the processes are hybrid. The present research is focused on the development and investigation of a combined method. The main idea of this combined method is to increase the depth of the residual compressive stress zone and also the depth of the zone where the microhardness is greater than that of the bulk (unaffected) material while preserving and expanding all known advantages of the S-phase in the surface layer.

There is very limited information on combined processes involving SCW aimed at further improving the rotating fatigue strength of CNASS specimens. The combination of smoothing DB and subsequent heat treatment (heating at 350 °C for three hours, followed by air-cooling) of 304 CNASS specimens increased the fatigue limit from 540 MPa (after DB) to 580 MPa, as has been achieved via hardening single-pass DB [3]. These authors explained the phenomenon of time-dependent diffusion-based strain-aging of the plasticized layers due to DB intervention. Lin et al. [45] sequentially implemented dynamic SCW, namely surface mechanical attrition treatment, followed by plasma LTN of 321 CNASS in order to increase the thickness and hardness of the nitrided layer as a result of improved nitrogen diffusion into the substrate. The improved SI is reflected in a significant increase in the wear resistance and load capacity of the nitrided layers of the corresponding CNASS component. Using the same combined approach, Chemkhi et al. [46] have significantly improved the SI of 316 CNASS coupon samples. The conducted literature survey shows that there is no information about the effect of the sequential application of the static SCW process and LTN on the fatigue strength of CNASS.

The aim of the present work is to develop a combined process based on sequentially applied DB and low-temperature gas nitriding (LTGN) and to investigate its effectiveness in improving the fatigue limit of CNASS.

## 2. Materials and Methods

AISI 304 CNASS, obtained as cylindrical hot-rolled bars, was from the same batch used in our previous studies [2,3]. All subsequent treatments were conducted on the material as received. The chemical composition (in terms of wt%), established by optical emission spectrometry, was as follows: Fe—71.5, C—0.036, Si—0.193, Mn—1.52, P—0.03, S—0.026, Cr—17.7, Ni—8.3, Mo—0.182, Cu—0.25, Nb—0.042, Ti—0.003, V—0.07, W—0.05 and others—balance. The basic mechanical characteristics, established at room temperature using a Zwick/Roell Vibrophore 100 testing machine to conduct tensile tests, were as follows: Young's modulus—198 GPa; yield limit—432 MPa; tensile strength—734 MPa and elongation—41%. A Bruker D8 Advance X-ray diffractometer, in conjunction with the Crystallography Open Database to determine the peak positions, was used to conduct the phase analysis. The microstructure and surface fracture were observed via scanning electron microscopy (SEM, LYRA I XMU Tescan, Brno, Czechia).

Specimens with dimensions of $\phi 20 \times 30$ mm were used to measure the physical-mechanical characteristics of SI. The surface microhardness measurements were made using a ZHVμ Zwick/Roell microhardness tester using a 0.015 kgf load and a holding time of 10 s. Twenty measurements were made for each specimen. The final value of the surface microhardness corresponded to the grouping center. The residual stresses were measured according to the methodology previously described [3].

DB implementation was described in our previous studies [2–4]. In this study, smoothing and single-pass hardening DB processes were implemented. Low-temperature gas nitriding (LTGN) was carried out using the experimental setup shown schematically in Figure 1. The LTGN installation was created at the Technical University of Varna by Y. Argirov. The laboratory installation consists of a vacuum furnace, a gas supply with gas flow regulation and a separate secured area for the storage of cylinders. The process is cyclical, with one working cycle lasting one hour at a working temperature of 420 °C

and under vacuum (0.7 μbar). Twenty working cycles were used for the LTGN of the specimens. To depassivate the surface layer of the samples, ammonium chloride was used as a depassivating agent in the furnace.

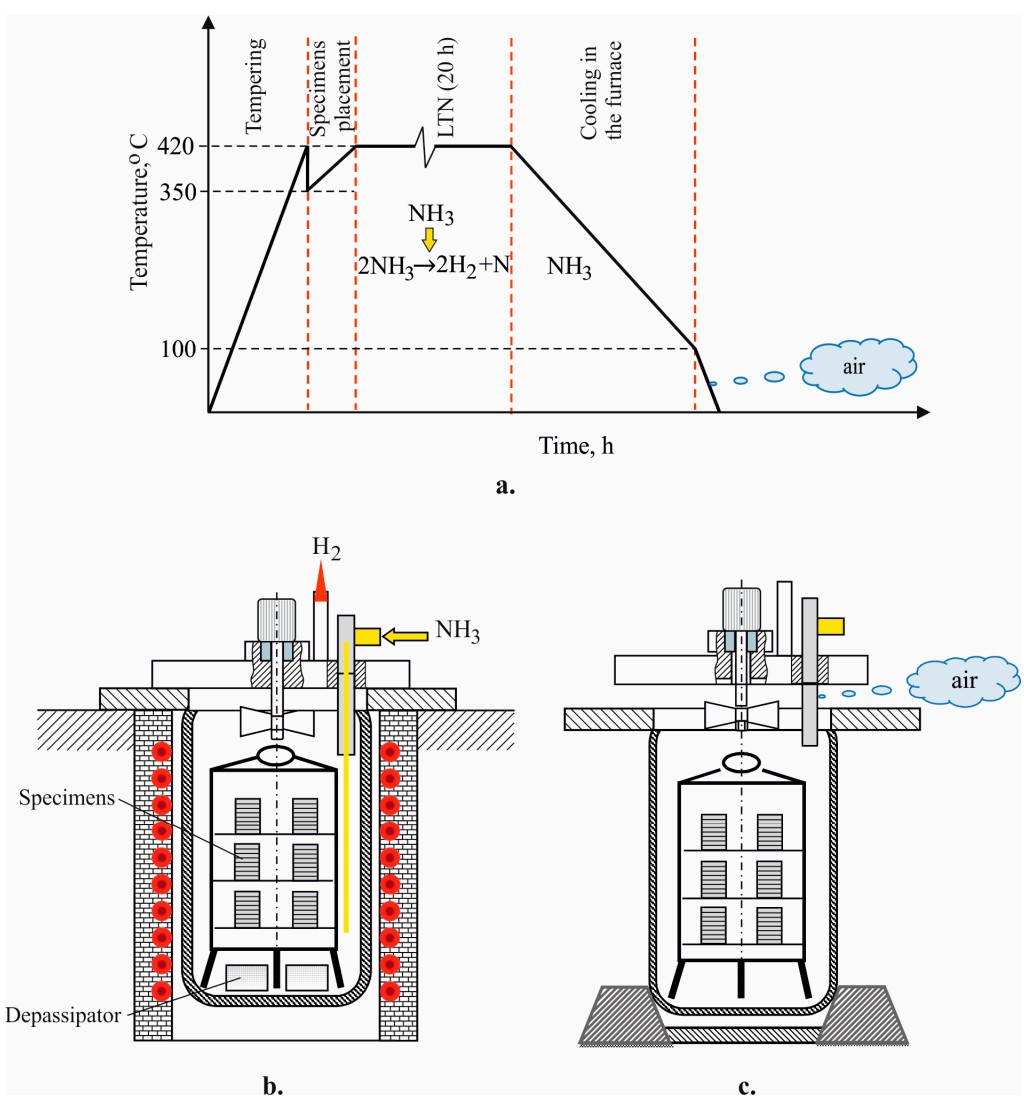

**Figure 1.** Scheme of the LTGN experimental setup: (**a**) cyclogram; (**b**) LTGN scheme in a shaft furnace and (**c**) air cooling.

Rotating bending fatigue tests were conducted on a UBM testing machine. The cycle asymmetry factor is R = −1. The loading frequency was 50 Hz in the air. The accuracy of counting the number of cycles to fatigue failure was 100 cycles. The experimental setup, methodology and fatigue specimen shape and size have been described [2,3]. The tests were conducted in two stages. In the first stage, samples treated using only different DB processes were tested. The fatigue outcomes previously obtained [2,3] were supplemented and significantly expanded. In the second stage, the fatigue tests were conducted with three groups of samples: turned, polished and gas nitrided (Group I); turned, smoothing diamond burnished and gas nitrided (Group II); and turned, hardening diamond burnished with one pass and gas nitrided (Group III). The fatigue specimens from Group I were polished before LTGN in order for the three groups of specimens to have approximately the same roughness and height parameters.

## 3. Results and Discussion

### 3.1. Effect of DB and LTGN on the Microstructure

The initial microstructure and its evolution due to DB intervention were studied previously [2,3]. In the present study, a comparison was made of the microstructures of the surface layers in correlation with the phase composition obtained after the following: (1) turning and LTGN; (2) smoothing DB and LTGN and (3) hardening DB and LTGN. Thus, the influence of the degree of plastic deformation (due to DB) and LTGN on the obtained strengthened layers was evaluated.

Phase analyses were performed on the cylindrical surfaces of the three specimens, each with dimensions of $\phi 20 \times 30$ mm, processed as previously stated. The established phases and their intensities are shown in Figure 2. The surface layers of the first two samples processed, i.e., after treatment options (1) and (2), besides a very high-intensity S-phase, also contained $\gamma$-phase and $\varepsilon$-phase (the chemical compound $Fe_{2-3}N$) in different proportions. The surface layer of the third sample (processed by hardening DB and LTGN) does not contain $\gamma$-phase. The severe surface plastic deformation introduced by hardening DB [2,3] is the reason for the transformation $\gamma \rightarrow \alpha'$. The intensity of the $\varepsilon$-phase is significant, while the S-phase is weakly expressed.

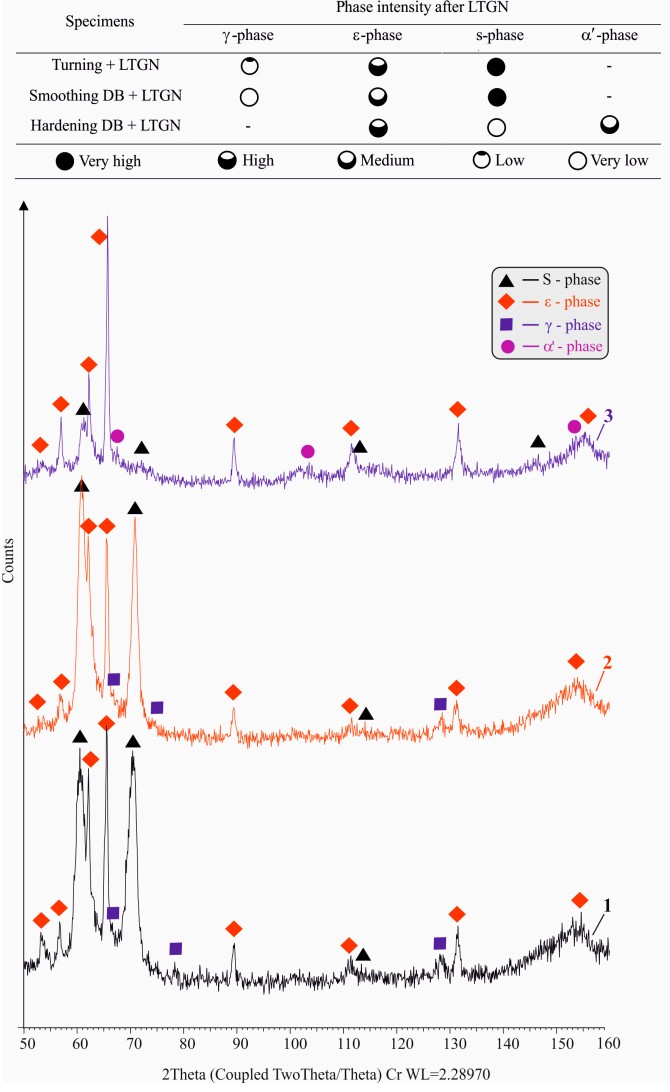

**Figure 2.** Phase analysis results: 1—turning and LTGN; 2—smoothing DB and LTGN and 3—hardening DB and LTGN.

Figure 3 shows the microstructure near the surface layer after turning, polishing and LTGN. In this study, the affected layer (AL) refers to the strengthened layer (nitrogen-rich layer) resulting from LTGN. The AL is 10–12 μm thick (Figure 3a). Based on the relatively large thickness, it can be assumed that the three established phases (S, $\varepsilon$ and $\gamma$) are in the form of a phase mixture in the AL. This means that nitrogen-bearing austenite with a $\gamma$-cubic lattice partially exists in this layer. A relatively large amount of $Fe_{2-3}N$ with a high degree of homogeneity was formed in the AL. Based on the (011) and (002) planes, the degree of tetragonality of the S-phase was determined: a = 2.655 Å, c = 4.286 Å and c/a = 1.6. The EDX analysis close to the boundary with the bulk material showed a local nitrogen content of 27.33 at% (Figure 3b). No nitrogen was detected under the AL, i.e., the structure is the same as before the LTGN intervention.

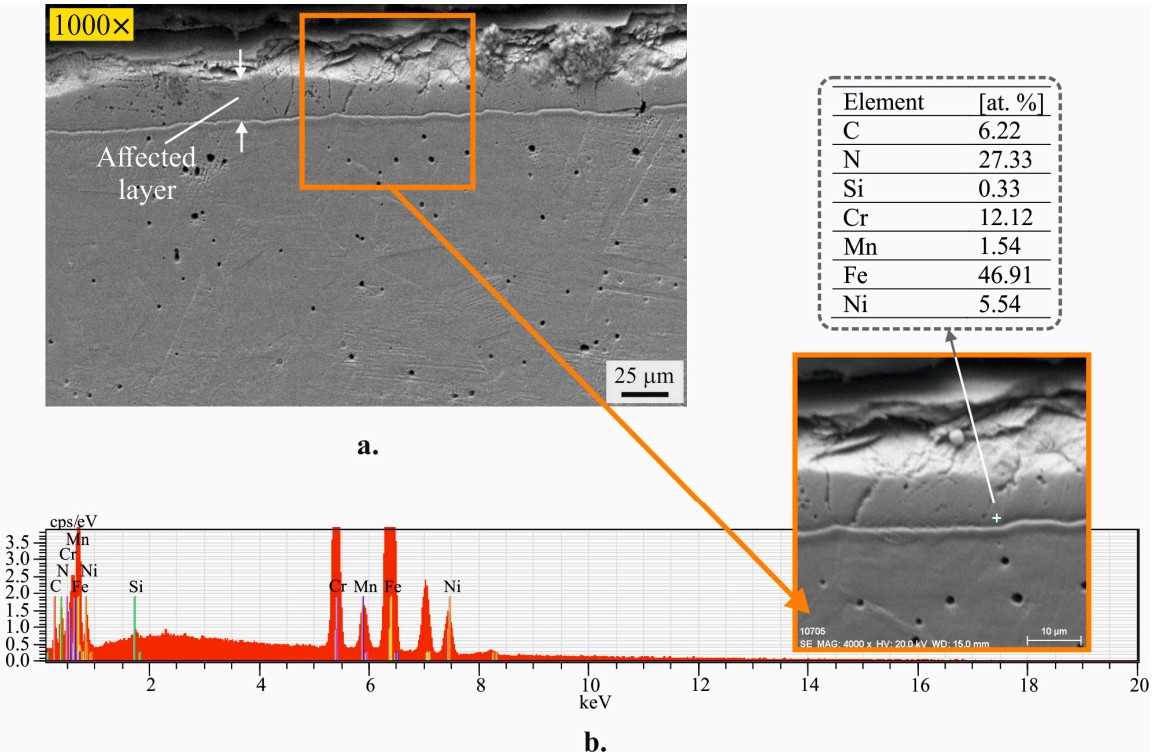

**Figure 3.** Microstructure near the surface layer after turning, polishing and LTGN: (**a**) affected layer; (**b**) EDX outcomes.

Figure 4 shows the microstructure near the surface layer after smoothing the DB and subsequent LTGN. Fine slip lines are observed below the AL, a consequence of the smoothing of the DB. The amount of the chemical compound, $Fe_{2-3}N$, in the AL is less compared to that of the first sample (see Figure 2). The local nitrogen content measured at this point is 17.2%. From the crystallographic planes (011) and (002) established by the X-ray diffractometer, the degree of tetragonality of the S-phase was determined (a = 2.7042 Å, c = 3.94508 Å and c/a = 1.458), less than that of the first sample.

Figure 5 shows the microstructure near the surface layer after hardening DB and subsequent LTGN. The structure under the AL has a well-defined deformation band (Figure 5a), a consequence of severe surface plastic deformation due to the hardening DB. There is no indication of recrystallization processes. The AL consists of S-phase and $\varepsilon$-phase with a high degree of homogeneity and strain-induced $\alpha'$-martensite introduced by hardening DB before LTGN. The technological temperature of 420 °C is not high enough for the reverse transformation, $\alpha' \rightarrow \gamma$, at which point the already formed martensite is stabilized by dissolving nitrogen in its crystal lattice. Thus, the $\gamma \rightarrow S$-phase transformation in the surface layer is hindered to some extent, confirmed by the fact that compared to the first two samples, the S-phase is of the lowest intensity. The local nitrogen content,

measured at a point, is 17.45 at% (Figure 5b) and is slightly higher than that of the second sample, but this is at the expense of the ε-phase, which is of greater intensity. The degree of tetragonality of the S-phase (a = 2.964 Å, c = 3.888 Å and c/a = 1.3117) is less than that of the first two samples.

Based on the results of the phase and microstructural analyses, the following conclusions can be drawn:

- The plastic deformation of the surface layer introduced before LTGN by static cold working with sliding friction contact (i.e., by DB) diminishes the diffusion process aiming at nitrogen enrichment of the surface layer and the closely located subsurface layers, which is confirmed by EDX analyses. Thus, as the degree of surface plastic deformation increases, the quality of the S-phase formed by LTGN deteriorates. The preliminary impact through severe surface plastic deformation reduces the intensity of the S-phase but increases the intensity of the ε-phase and leads to the appearance of a new (also hard) phase: stabilized nitrogen-bearing martensite.
- This conclusion is generally supported by Proust et al. [47], who, for 316L steel, found that the average depth of the nitrogen-rich layer is 24.8 μm (425 °C/20 h plasma nitriding), but after preliminary surface mechanical attrition treatment, it is only 5 μm.
- Ignoring the negligible amount of γ-phase in the first two samples, the ALs of all three samples contain hard phases in different proportions. A significant increase in microhardness and surface residual stresses can be anticipated for each of the three processes: (1) turning and LTGN; (2) smoothing DB and LTGN and (3) hardening DB and LTGN.

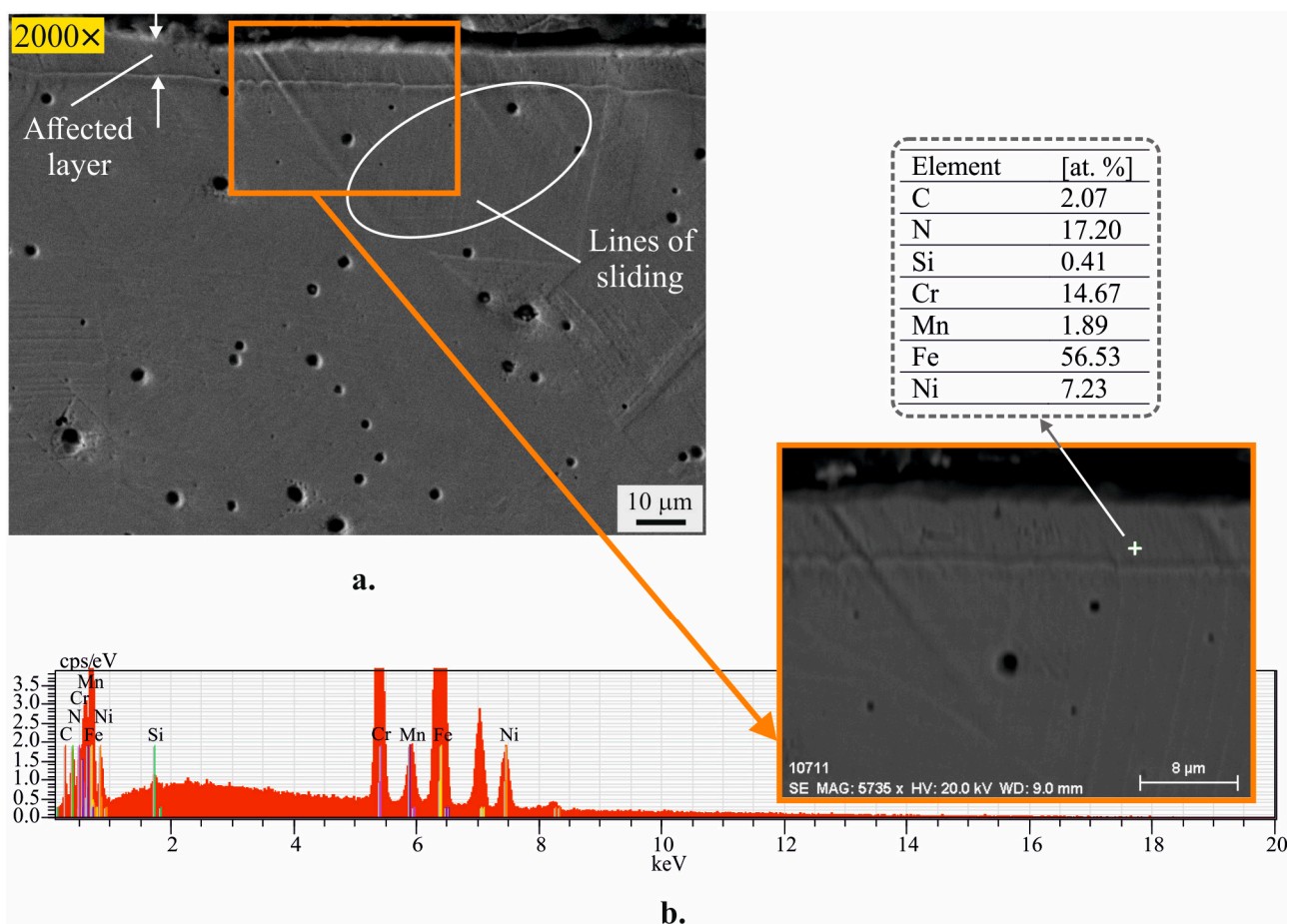

**Figure 4.** Microstructure near the surface layer after LTGN and smoothing DB: (**a**) affected layer; (**b**) EDX outcomes.

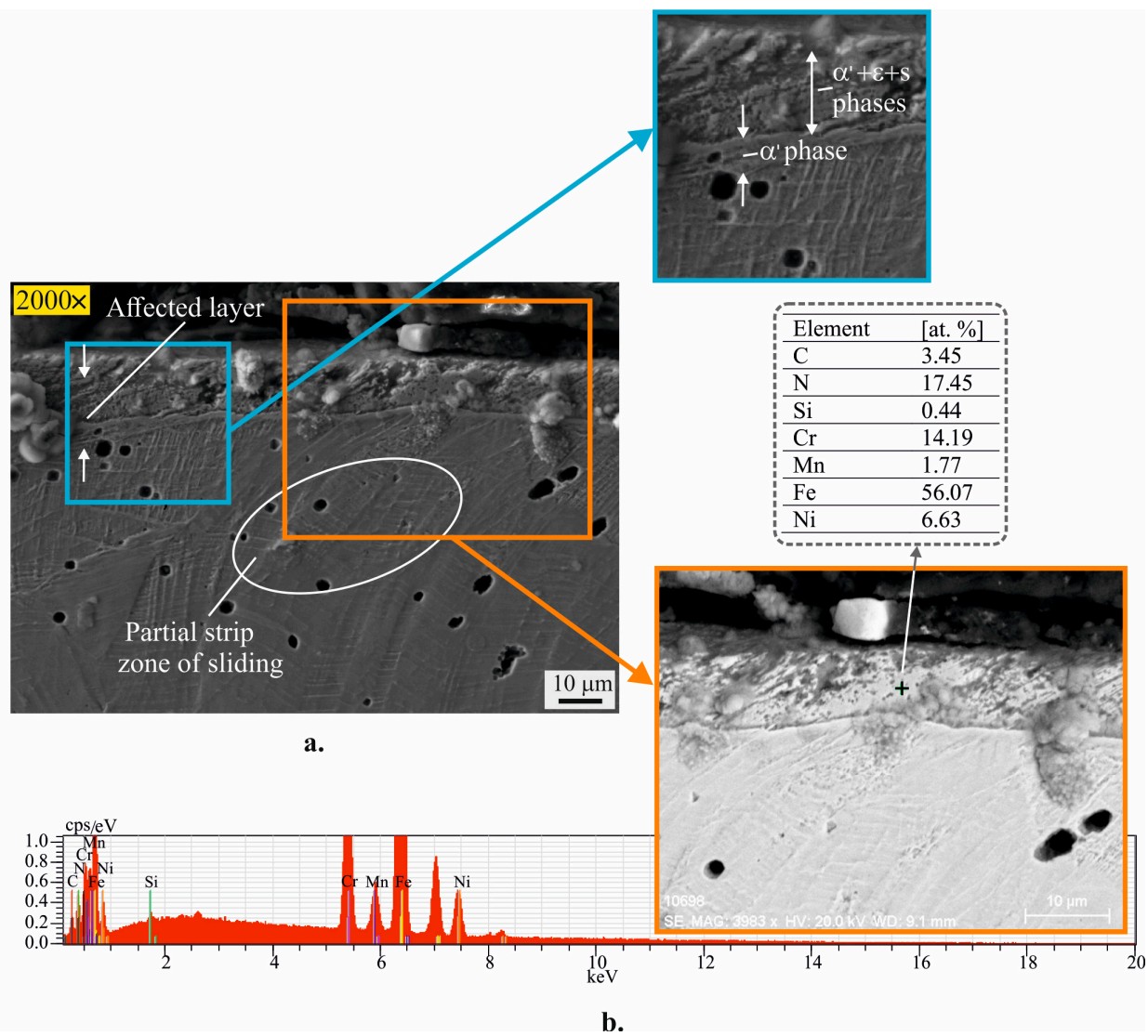

**Figure 5.** Microstructure near the surface layer after LTGN and hardening DB: (**a**) affected layer; (**b**) EDX outcomes.

### 3.2. Effect of DB and LTGN on the SI Physical-Mechanical Characteristics

#### 3.2.1. Microhardness

Figure 6a shows the microhardness profiles before LTGN. The microhardness of the bulk material is approximately 420 HV. Turning slightly increases the microhardness at a depth of approximately 0.07 mm. The subsequent DB significantly increases the microhardness, reaching 575 HV at a depth of approximately 0.01 mm in both variants (smoothing or hardening). However, while the depth of the hardened layer due to smoothing DB is below 0.1 mm, hardening DB achieves a significantly greater depth. Figure 6b shows the microhardness profiles after LTGN, with the prior treatments being, respectively, turning, smoothing DB, and hardening DB. The highest microhardness is provided by LTGN after turning—approximately 2000 HV at a depth of several micrometers—and the depth of the hardened layer is only 0.03 mm. When the prior treatment is DB (either smoothing or hardening), the effect of LTGN on the microhardness decreases with respect to the near-surface layers: the maximum microhardness at a depth of 5 μm is 1400 HV and 1700 HV, respectively. At the same time, the depth of the hardened layer is greater, a consequence of the cold work introduced through DB. The explanation of this phenomenon is that the presence of severe surface plastic deformation introduced through DB prevents the

formation of a quality S-phase, which is the hardest compared to the other established phases in the ALs.

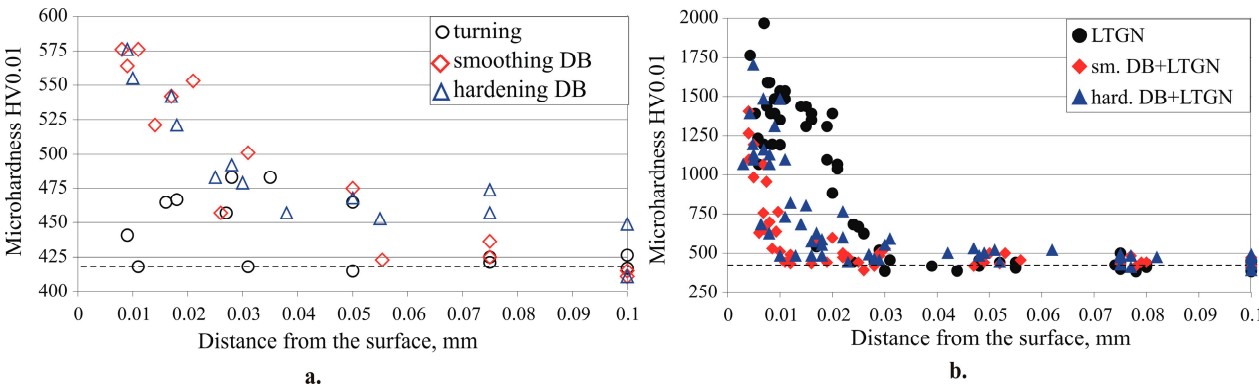

**Figure 6.** Microhardness profile: (**a**) before LTGN; (**b**) after LTGN.

### 3.2.2. Residual Stresses

As found previously [3], single-pass hardening DB introduces about 50% strain-induced $\alpha'$-martensite into the surface layer, which quickly decreases to zero at a depth of up to 0.1 mm [2]. Therefore, in the present study, the residual stresses were measured for the austenite phase.

The distribution of residual stresses introduced by turning, smoothing DB and hardening DB, i.e., before LTGN, is shown in Figure 7. The turning introduces tensile residual stresses (axial and hoop) in the surface layer. Conversely, both DB variants, smoothing and hardening, introduce significant residual compressive stresses. The axial residual stresses are significantly larger in absolute value compared to the hoop stresses at a depth of approximately 0.1 mm, after which they decrease to zero (in contrast to the hoop stresses). As can be expected, the hardening DB introduces larger compressive residual stresses in the surface and the closely spaced subsurface layers.

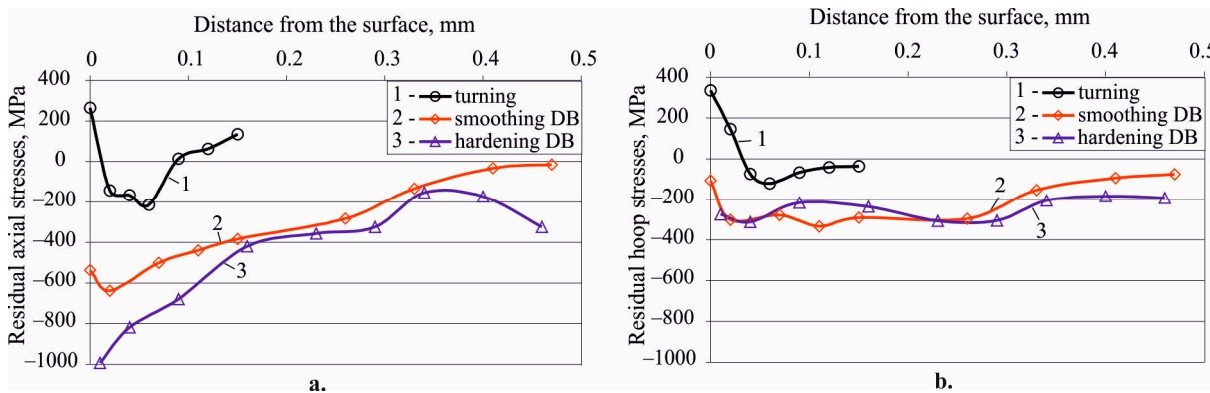

**Figure 7.** Residual stresses before LTGN (1—turning; 2—smoothing DB and 3—hardening DB): (**a**) axial; (**b**) hoop.

Figure 8 depicts the residual stress distribution after LTGN and following different initial treatments (turning, smoothing DB or hardening DB). LTGN (after turning and polishing) introduces maximum residual compressive stresses in the surface layer and in the closely spaced subsurface layers at a depth of up to 10 μm. However, the compressive zone depth is only 50 μm. The surface residual stresses introduced by the static SCW method on external cylindrical surfaces are characterized by the fact that the axial surface stresses are much larger in absolute value than the surface hoop stresses [2,3]. However, the surface axial stresses introduced through LTGN are 2.43 times lower than the hoop stresses,

whose absolute value reaches 6.6 GPa. The combination of smoothing DB and subsequent LTGN introduces compressive residual stresses at 6 times greater depth (0.3 mm) compared to turning and LTGN, but the surface axial and hoop residual stresses are smaller in absolute value, at 2.1 and 3.94 GPa, respectively. However, the difference between the surface axial and hoop stresses decreases, with the axial stresses being 1.88 times lower than the hoop stresses. The combination of hardening DB and subsequent LTGN intensified the observed trend. The surface residual axial and hoop stresses are even smaller (1.12 and 1.22 GPa, respectively), and the compression zone increases its depth to more than 0.4 mm. The surface axial and hoop stresses are almost equal.

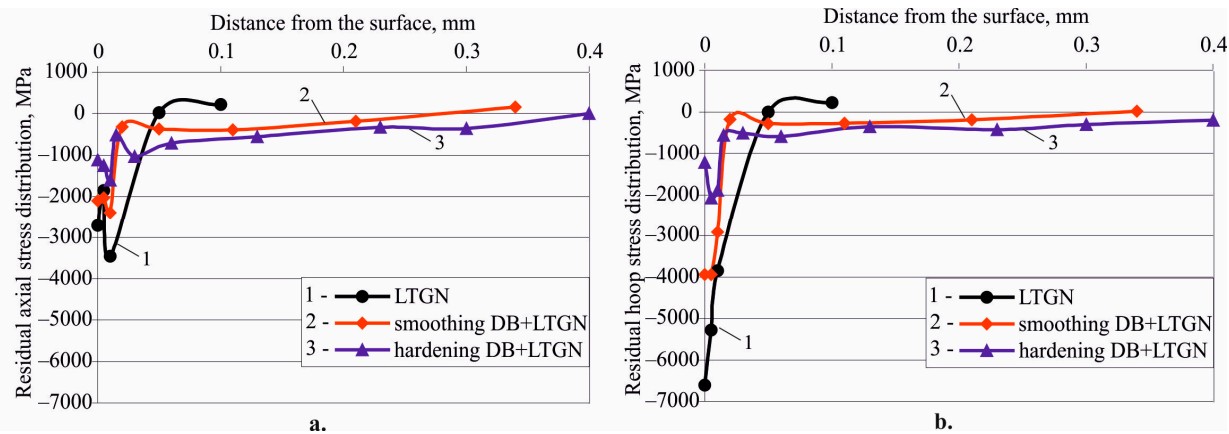

**Figure 8.** Residual stress distribution after LTGN at different initial conditions (1—turning + LTGN; 2—smoothing DB + LTGN and 3—hardening DB + LTGN): (**a**) axial; (**b**) hoop.

The comparison between Figures 7 and 8 shows the final result of the evolution of the residual stresses introduced by smoothing and hardening DB, and this is illustrated in Figure 9. The chemical-thermal impact during LTGN simultaneously causes two effects: (1) relaxation of residual compressive stresses introduced by surface cold work that occurs at a depth greater than 0.02 mm; (2) occurrence of maximum surface residual compressive stresses at a depth of up to 10 μm as a result of the formed new phases having an increased volume of crystal lattice in the AL. The second effect is mainly due to the S-phase formed. The S-phase has the lowest intensity after hardening DB. Therefore, the residual compressive stresses in the AL are significantly smaller (Figure 9b) compared to smoothing DB (Figure 9a).

To evaluate the effect of only the thermal impact when heating up to 420 °C as a function of time, Figure 10 shows the alteration of the residual stresses introduced by smoothing and hardening DB. The thermal impact of 420 °C for 20 h causes a relaxation of the residual compressive stresses introduced through the DB and, as a consequence, a reduction of the depth of the compressive zone (Figure 10). At the same time, as a consequence of the hard phases formed (S, ε and stabilized nitrogen-bearing martensite), the chemical, physical and mechanical properties of the surface layers at a depth of several micrometers change drastically. These two factors (the technological temperature of 420 °C and the new hard phases formed), acting simultaneously and interdependently, are the cause of the residual stress evolution.

It can be assumed that regardless of the smaller surface residual stresses, the reduced gradient of the residual stress distribution and the deeper compressive zone obtained by the two combined DB and LTGN processes will improve the fatigue behavior to a greater extent than that achieved by the turning and LTGN processes.

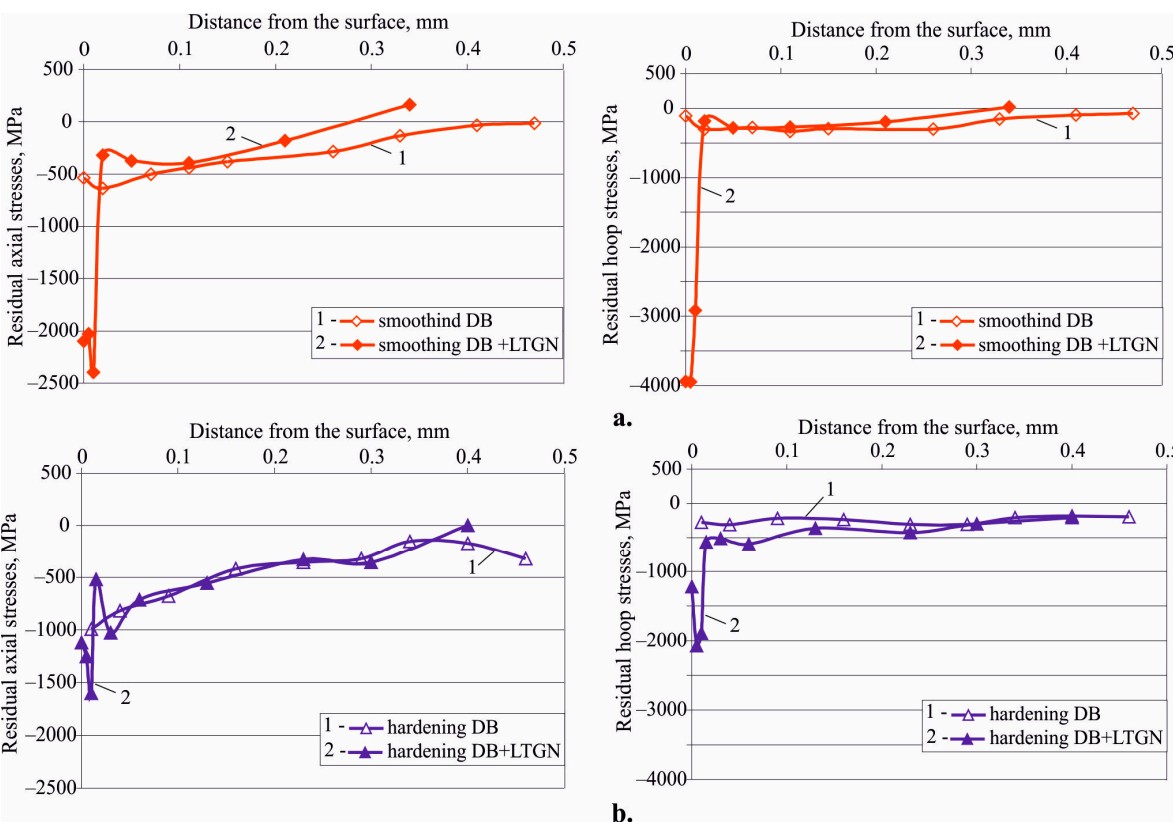

**Figure 9.** Evolution of the residual stresses introduced through DB as a result of the LTGN process: (**a**) smoothing DB; (**b**) hardening DB.

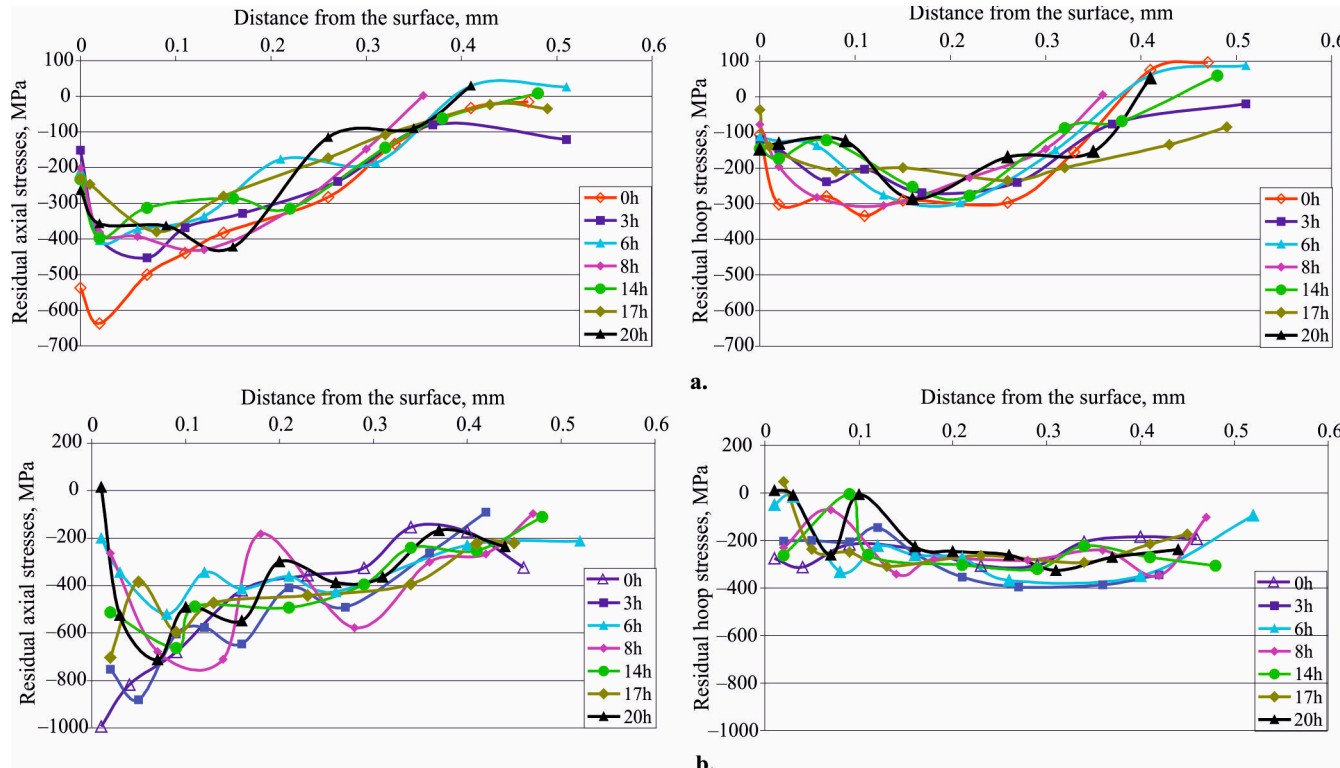

**Figure 10.** Relaxation of the residual stresses introduced through DB when heated to 420 °C as a function of time: (**a**) smoothing; (**b**) hardening.

*3.3. Effect of DB and LTGN on Fatigue Behavior of 304 CNASS*

3.3.1. S-N Curves

Figure 11 shows the S-N curves of specimens treated with different types of DB. The reference condition is the S-N curve obtained for samples processed only by turning and polishing, which leads to a fatigue limit of 440 MPa. Smoothing DB increases the fatigue limit to 540 MPa, i.e., an improvement of 22.7% compared to the reference samples. If the samples after smoothing DB are heated to 350 °C for 3 h, the fatigue limit increases to 580 MPa, and the improvement compared to the base level is 31.8%. The reason for this improvement is the time-dependent diffusion-based strain-aging [3]. The same improvement of 31.8% is achieved by single-pass hardening DB. The highest fatigue limit of 605 MPa is achieved by five-pass hardening DB, an improvement of 37.5% compared to the reference samples. However, the hardening DB creates a significant plastic deformation of the surface layer, which is the cause of the $\gamma \to \alpha$ transformation. The strain-induced $\alpha'$-martensite increases the surface microhardness but decreases resistance to electrochemical corrosion [3].

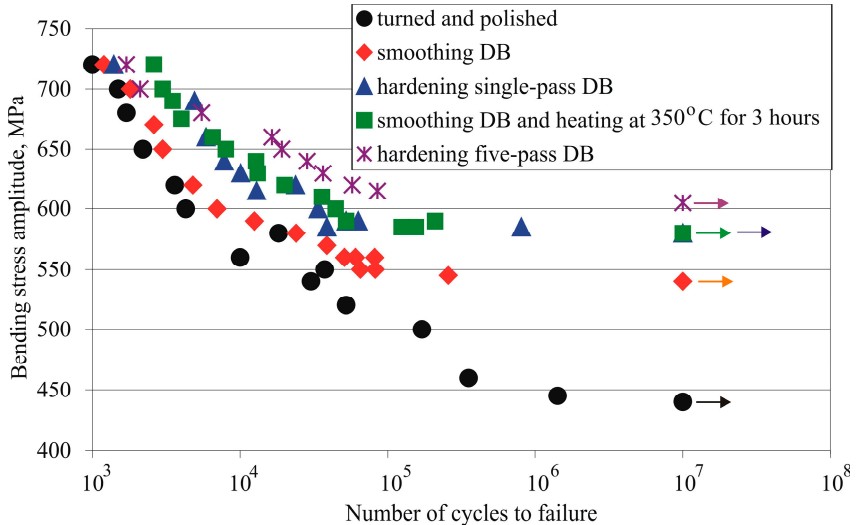

**Figure 11.** S-N curves of specimens treated via different types of DB.

Figure 12 shows the S-N curves of the three groups of specimens treated via LTGN after different pretreatments. The pretreatment for Group I samples increases the fatigue limit from 440 MPa (reference point) to 580 MPa, i.e., an improvement of 31.8%, which was achieved by smoothing DB followed by heating up to 350 °C for 3 h as well as by single-pass hardening DB. Groups II and III specimens achieve the same fatigue limit of 600 MPa, i.e., an improvement over the reference value by 36.3%. The improvement compared to Group I is 3.45%. However, the pretreatment of the Group II and Group III specimens increases the fatigue life more than 178 times compared to the turning and LTGN processes. The main reason for this improvement is the smaller residual stress distribution gradients and the deeper compressive zones obtained by the two combined DB and LTGN processes. The factors determining the comparable fatigue limit of 600 MPa for the two combined processes (smoothing DB and LTGN and single-pass hardening DB and LTGN) are of interest. In the literature, the improvement of the fatigue behavior of austenitic steels after LTN is primarily associated with the quality of the formed S-phase in the surface layer [36]. The obtained equal fatigue limit for the two combined processes based on DB and LTGN facilitates the explanation of the following conclusion. The fatigue behavior depends on the combination of SI characteristics obtained, including the types of phases and their ratio in the AL on the one hand and the profile of residual stresses on the other. The resulting combination of SI characteristics is the result of the superimposition of strain hardening followed by transformation hardening (resulting in the formation of new hard

phases with an increased volume of crystal lattice). The presence of a higher-intensity S-phase in the AL and the creation of larger surface compressive residual stresses are not, by themselves, dominant factors in raising the fatigue limit. Evidence of this is provided by the fact that the fatigue limit obtained for the combined process based on hardening DB and LTGN results in an S-phase of lower intensity and smaller compressive axial and hoop surface stresses. At the same time, however, this process provides the hard phase ε and stabilized nitrogen-bearing martensite with greater intensity (Figure 2), as well as a greater depth of the compressive zone (Figure 9). Therefore, the combined processes based on an appropriate combination of static surface cold work and LTN parameters can provide a synergistic effect to improve the fatigue behavior of CNASS.

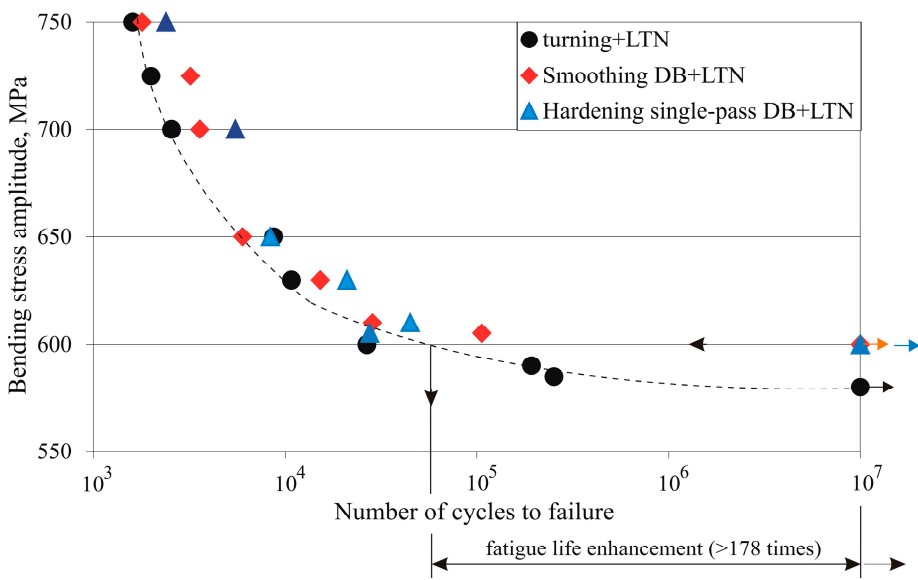

**Figure 12.** S-N curves of specimens subjected to LTGN after different prior treatments.

### 3.3.2. Fractography

Figure 13 shows the fracture surface of the fatigue specimen from Group I (stress amplitude of 590 MPa; number of cycles to failure: 250,300). Several centers of macrocrack formation are observed, but only the one outlined in red is a major macrocrack (Figure 13a). This crack did not necessarily form first. It is possible that before this, another macrocrack formed and developed, but split into several sections upon encountering a local obstacle. As a result, the local stress, which is the cause of macrocrack growth, sharply decreases below its critical value, whereupon the development of another macrocrack begins.

The mechanism of formation and development of the main macrocrack is shown in Figure 13b. Region A shows the nucleation of microcracks that merge and form the macrocrack. These microcracks are localized in the microzone of opposing slip systems, causing displacement of dislocations in two intersecting planes. This microzone is situated at the boundary between the hard, but brittle, AL and the bulk material. Thus, the microzone is a natural stress concentrator and the formation site of the macrocrack that develops and causes a transcrystalline crack in the AL. The main macrocrack formed develops under the action of critical local stress between planes, with a different direction of movement of the cyclic wave (zone 1 and zone 2 in Figure 13b—general view). Zone 1 has weak slip lines, and zone 2 has an intercrystalline crack formed under the condition of hindered deformation. On both sides of the main macrocrack, linear cracks (regions B and C) are observed, as well as a region with extrusion, which is characteristic of plasticity due to high local stresses. On this basis, it can be assumed that the development of the macrocrack has a rather plastic character.

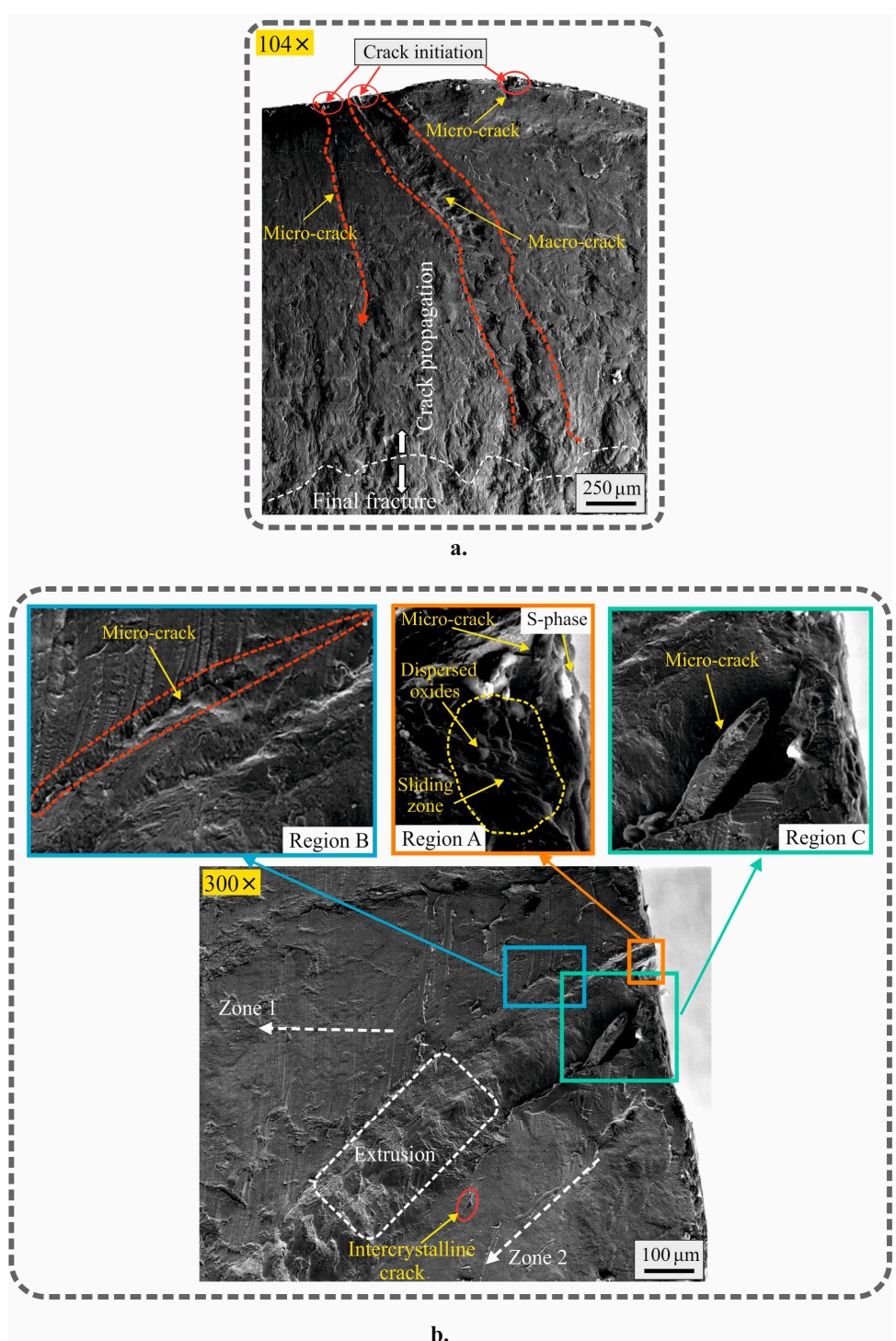

**Figure 13.** Fracture surface of the fatigue specimen (stress amplitude of 590 MPa; number of cycles to failure: 250,300), treated by turning + LTGN: (**a**) main macrocrack; (**b**) mechanism of formation and growth of main macrocrack.

Figure 14 shows the fracture surface of the fatigue specimen from Group II (stress amplitude of 605 MPa; number of cycles to failure: 106,200). In Figure 14a, three zones are outlined. The micro-area of nucleation of the microcracks and macrocrack formation is located at the border between the AL and the bulk material in zone 1. The direction of the cyclic wave differs by a small angle from the radial direction. At the boundary between zones 1 and 2, an extrusion region is formed, almost perpendicular to the cyclic wavefront. The extrusion region exerts a strong resistance to the propagation of the cyclic wave and

greatly increases the local alternating stresses. The resistance to the cyclic wave increases, and a static boundary zone forms that separates zone 2 from zone 3. As a consequence of the hindered development of the cyclic wave (leading to a sharp increase in local stresses), grain cracks (intercrystalline destruction) are formed in zones 2 and 3. The shape of the grain boundary cracks gives an indication of the tough-plastic behavior of the material. One of the prerequisites for the formation of the three zones is the linear arrangement of dislocations on separate polygonization planes. The arrangement of dislocations along grain boundaries is a prerequisite for intercrystalline destruction.

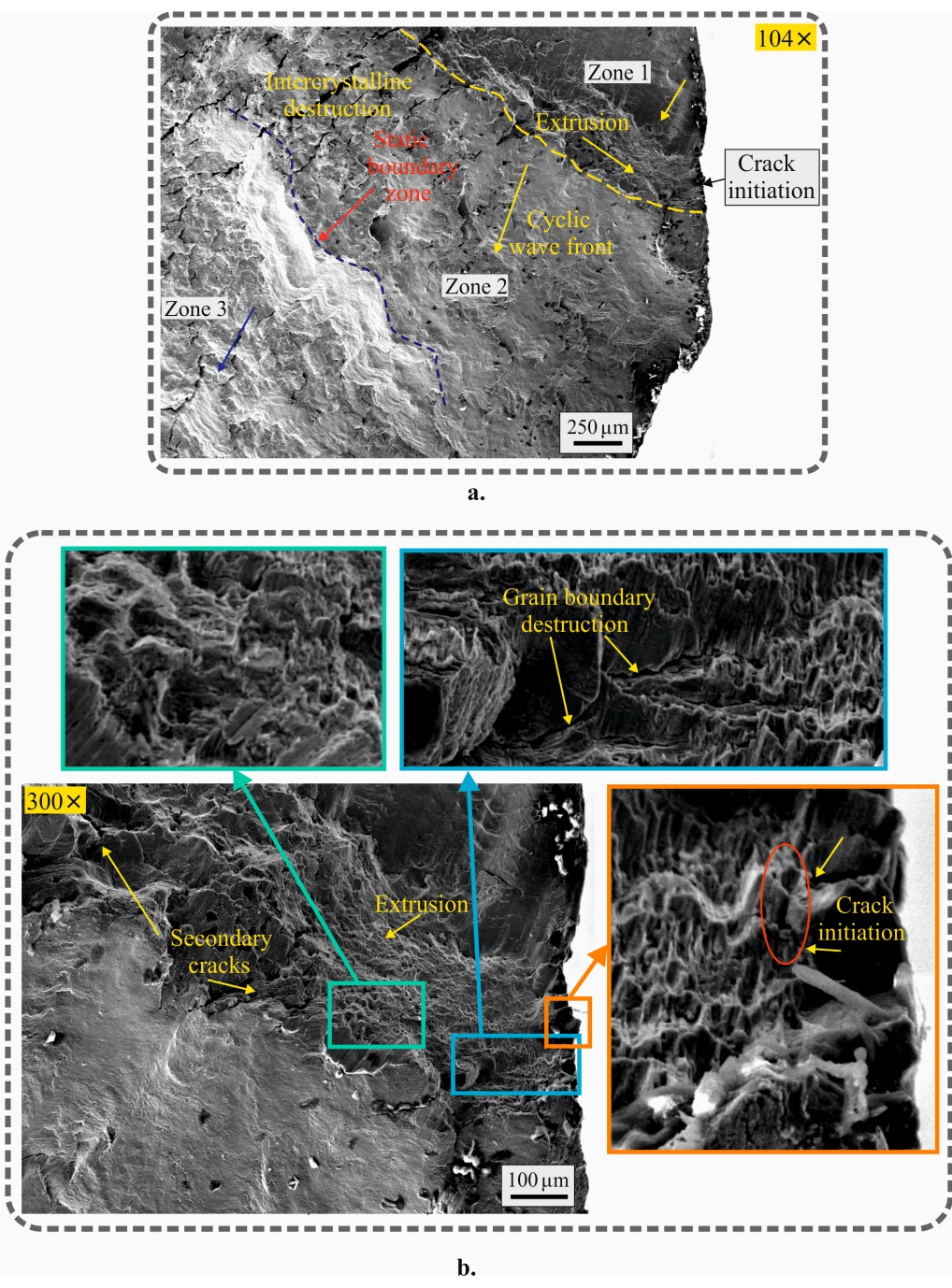

**Figure 14.** Fracture surface of the fatigue specimen (stress amplitude of 605 MPa; number of cycles to failure: 106,200), treated by smoothing DB + LTGN: (**a**) general view; (**b**) mechanism of formation and growth of main macrocrack.

The mechanism of macrocrack formation and development (Figure 14b) is similar to that of the specimen in Figure 13, but the crack-developing mechanism is more pronounced. The material below the AL shows tough-plastic behavior, and the destruction has a pronounced plastic character.

Figure 15 shows the fracture surface of the fatigue specimen from Group III (stress amplitude of 610 MPa; number of cycles to failure: 78,200). The center of macrocrack formation and two main zones are clearly distinguishable (Figure 15a). Zone 1 shows the inhomogeneous movement of the cyclic wave (indicated by red arrows), creating a condition for extrusion. In zone 2, the cyclic waves also have different directions. As a result, static local regions are formed, which create local supercritical stresses. In these static regions of impeded plastic deformation, intercrystalline cracks are observed (Figure 15b). In zone 1, the cracks are smaller, but mixed-type (inter- and transcrystalline) cracks are also observed.

The mechanism of macrocrack formation and development differs to some extent compared to the specimen in Figure 14. The initial stage of crack formation involves sliding along different crystallographic planes and an increase in dislocation density around the intersection of the two planes. The hardening of DB significantly increases the dislocation density along the favorable grain polygons formed. Cyclic loading causes alternating stresses. When the tensile stresses reach a limit value, they cause inter-boundary rupture of the bonds between the grains. The macrocrack formed at the boundary between the AL and bulk material causes a transcrystalline crack in the AL. In the vicinity of the formed macrocrack, a static zone is observed, resulting from the contact of cyclic wave A with cyclic wave B at right angles to each other. The meeting of the two waves is unfavorable and causes the formation of both trans- and intercrystalline cracks. In general, it can be assumed that the destruction has a plastic character, regardless of the fact that in individual local areas, it is hybrid.

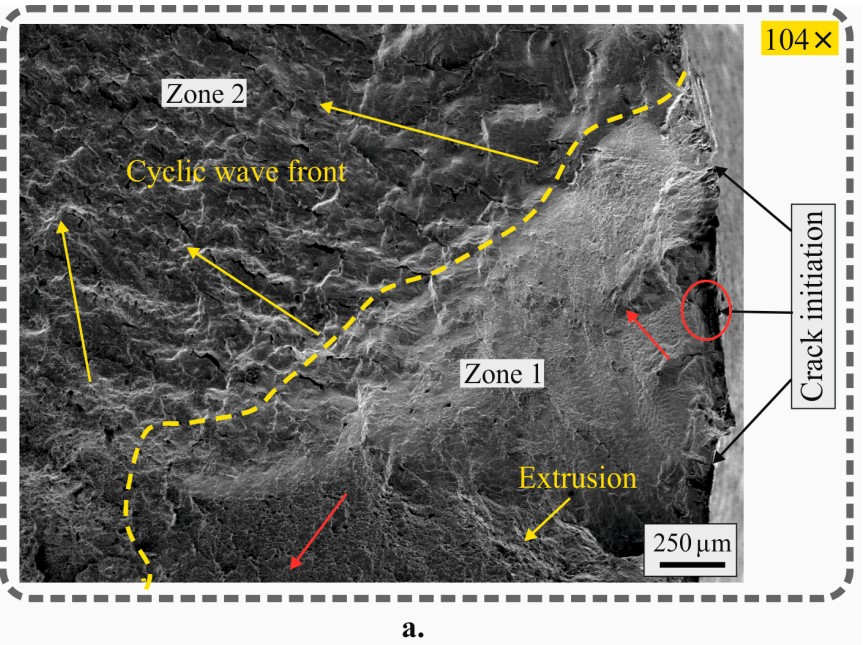

**Figure 15.** *Cont.*

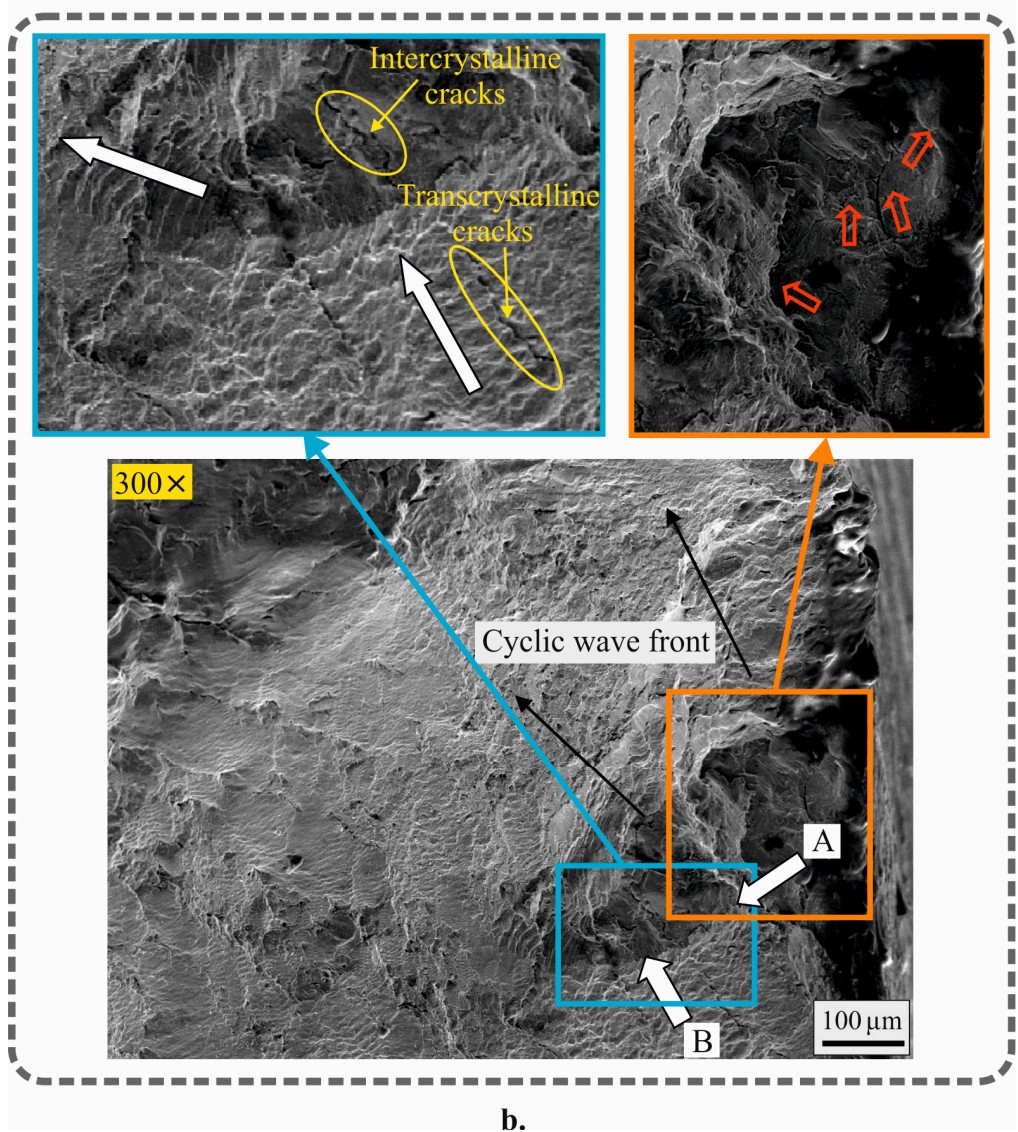

**Figure 15.** Fracture surface of the fatigue specimen (stress amplitude of 610 MPa; number of cycles to failure: 78,200), treated by hardening DB + LTGN: (**a**) general view; (**b**) mechanism of formation and growth of main macrocrack.

## 4. Conclusions

A new combined process based on sequentially applied DB and LTGN to optimally improve the fatigue strength of 304 steel has been developed. The new process achieves a fatigue limit of 600 MPa, an improvement of 36.4% compared to untreated specimens. The major new findings concerning the correlation "new process–surface integrity–fatigue behavior" are as follows:

- The approach generates a synergistic effect, as the obtained fatigue limit (600 MPa) is greater than that achieved by DB (540 MPa for smoothing or 580 MPa for single-pass hardening processes) or LTGN (580 MPa) alone. The synergistic effect is the result of the superimposition of two consecutive effects—strain hardening via DB and transformation hardening via LTGN—as a result of the formation of new solid phases with an increased volume of the crystal lattice. It is advisable to use smoothing DB (instead of hardening) in combination with LTGN, as lower roughness height parameters are obtained.
- As the degree of plastic deformation of the surface layer (introduced by DB) increases, the content of the S-phase in the nitrogen-rich layer formed by LTGN decreases, with

a resultant increased content of the $\varepsilon$-phase and a new (also hard) phase: stabilized nitrogen-bearing martensite.

- It was found that axial residual stresses introduced into external cylindrical surfaces by LTGN are 2.43 times smaller compared to the hoop stresses, whose absolute value reaches 6.6 GPa. This trend is opposite to the typical nature of the distribution of the two types of residual stresses after the static SCW method.
- The application of a combined process, including DB and subsequent LTGN, results in a much greater depth of zone with compressive residual stresses and significant surface axial and hoop residual stresses compared to turning and LTGN. At the same time, the hoop surface residual stresses decrease at the expense of the axial ones. These trends are more pronounced with an increase in the degree of preliminary plastic deformation.
- The two combined processes (smoothing DB and LTGN and single-pass hardening DB and LTGN) were found to achieve the same fatigue limit of 600 MPa, an improvement of 3.45% compared to the turning and LTGN processes. However, the two combined processes increase the fatigue life more than 178 times compared to the turning and LTGN processes. The main reason for this improvement is the reduced residual stress distribution gradient and deeper compression zone obtained by these processes.
- The formation of a fatigue macrocrack for all three groups of samples (turning + polishing + LTGN, smoothing DB + LTGN and single-pass hardening DB + LTGN) in the high-cycle fatigue field takes place at the boundary between the nitrogen-rich layer and the bulk material, which determines the similar mechanism of destruction of the three samples.

**Author Contributions:** Conceptualization, J.M. and G.D.; methodology, J.M. and G.D.; software, J.M. and G.D.; validation, J.M. and G.D.; formal analysis, J.M., G.D. and Y.A.; investigation, A.A., V.D., Y.A., G.D. and J.M.; resources, J.M. and G.D.; data curation, J.M. and G.D.; writing—original draft preparation, J.M. and G.D.; writing—review and editing, J.M. and G.D.; visualization, J.M., G.D., A.A. and V.D.; supervision, J.M.; project administration, J.M. and G.D.; funding acquisition, J.M. and G.D. All authors have read and agreed to the published version of the manuscript.

**Funding:** This research was funded by the European Regional Development Fund within the OP. "Science and Education for Smart Growth 2014–2020" and the Project CoC "Smart Mechatronics, Eco- and Energy Saving Systems and Technologies", No. BG05M2OP001-1.002-0023.

**Institutional Review Board Statement:** Not applicable.

**Informed Consent Statement:** Not applicable.

**Data Availability Statement:** Data are contained within the article.

**Conflicts of Interest:** The authors declare no conflicts of interest.

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
