# Peer review of "Improvement in Fatigue Strength of Chromium–Nickel Austenitic Stainless Steels via Diamond Burnishing and Subsequent Low-Temperature Gas Nitriding"

_applsci, doi:10.3390/app14031020_

Round 1

Reviewer 1 Report

Comments and Suggestions for Authors

The article entitled "Improvement in Fatigue Strength of Chromium-Nickel Austenitic Stainless Steels via Diamond Burnishing and Subsequent Low-Temperature Gas Nitriding" is written very comprehensively and follows on from previous research by the authors. The article is clearly divided into 4 parts, which logically follow each other. The research is carried out methodologically correctly and the measured results are presented in a logical sequence. The cited literature is in good connection with the content of the article.

I have reservations only about small things such as not specifying the necessary data of the UBM device on the line. At the same time, the designation of the device in which the nitriding was carried out is not indicated. The authors further complicate access to information by not describing the methods and equipment in the article and referring to previous articles. Figure 9 is a wrong description of figure a) and at the same time the designation of figure 10 is in the alphabet. I have other reservations about the information presented in the graphs, they do not logically follow each other, and it is difficult to understand so many of the processes presented, especially why the residual stress values are given in Figure 10 and how they relate to the cell.

Picture 3 shows a very cracked and chipped affected layer. Can the authors explain the metallographic procedure?

After correcting the mentioned errors, I recommend the article to be published in the journal.

Reviewer 2 Report

Comments and Suggestions for Authors

It is a correct scientific work about the fatigue strength of chromium-nickel austenitic stainless steel. The main objective is improving the fatigue strenght in these steels via diamond burnishing and subsequent low-temperature gas nitriding.

1.       XRD analysis: I recommend improving the XRD analysis. It is necessary to calculate the percentage of all phase detected in the XRD diffraction patterns.

2.       It is also necessary to calculate the crystallographic parameters. Options are: Williamson-Hall method, Rietveld refinement, ….

3.       Mechanical properties. I recommend adding statistical information about the data collected. I assume several experiments

4.       S-N curves: I suggest adding a fitting equation to the experimental data to check minor differences in the three specimens analyzed.

I also recommend minor English revision.

Comments on the Quality of English Language

I also recommend minor English revision.

Reviewer 3 Report

Comments and Suggestions for Authors The article describes in a very interesting way the fatigue behavior of nitrided austenitic stainless steel at low temperatures. It can be pointed out that the scientific support and development were well established in the continuity of previous work, so it may be appropriate for publication in this journal after justify the effect of roughness on fatigue tests after diamon burnishing and after nitriding. It can only be pointed out that an analysis of the structure of the nitriding and the  its correlation with the crack advance in the fractures after the fatigue tests would be very adequate in this contribution.

Round 2

Reviewer 2 Report

Comments and Suggestions for Authors

The authors modify the manuscript taking into account the comments of the reviewers.

The quality and soundness of the manuscript has been improved.

Comments on the Quality of English Language

 Minor editing of English language required